# High-Precision Inversion of Shallow Bathymetry under Complex Hydrographic Conditions Using VGG19—A Case Study of the Taiwan Banks

**Jiaxin Cui [1,2], Xiaowen Luo [1,2,*], Ziyin Wu [2], Jieqiong Zhou [2], Hongyang Wan [2], Xiaolun Chen [2] and Xiaoming Qin [2]**

[1] Institute of Sedimentary Geology, Chengdu University of Technology, No. 1 East Third Road, Erxianqiao Street, Chengdu 610059, China

[2] Key Laboratory of Submarine Geosciences, Second Institute of Oceanography, Ministry of Natural Resources, 36 North Baochu Road, Hangzhou 310012, China

*   Correspondence: luoxiaowen@sio.org.cn

**Abstract:** Shallow bathymetry is important for ocean exploration, and the development of high-precision bathymetry inversion methods, especially for shallow waters with poor quality, is a major research aim. Synthetic aperture radar (SAR) image data benefit from a wide coverage, high measurement density, rapidity, and low consumption but are limited by low accuracy. Alternatively, multibeam data have low coverage and are difficult to obtain but have a high measurement accuracy. In this paper, taking advantage of the complementary properties, we use SAR image data as the content map and multibeam images as the migrated style map, applying the VGG19 neural network (optimizing the loss function formula) for bathymetric inversion. The model was universal and highly accurate for bathymetric inversion of shallow marine areas, such as turbid water in Taiwan. There was a strong correlation between bathymetric inversion data and measured data ($R^2 = 0.8822$; RMSE = 1.86 m). The relative error was refined by 9.22% over those of previous studies. Values for different bathymetric regions were extremely correlated in the region of 20–40 m. The newly developed approach is highly accurate over 20 m in the open ocean, providing an efficient, precise shallow bathymetry inversion method for complex hydrographic conditions.

**Keywords:** shallow bathymetry inversion; VGG model; synthetic aperture radar image data; multibeam sonar data

## 1. Introduction

The determination of shallow sea bathymetry and underwater topography is an important part of ocean exploration. In particular, shallow sea bathymetry and underwater topography have an important reference value for mineral resource exploration, maritime rescue, maritime transportation, maritime military, maritime cash crop planting, and environmental protection. The shallow sea mainly refers to the flat and shallow seas with depths less than 20 m, and the shallow sea bathymetry within 50 m has an important role in studying marine geology, corals, and other marine life distribution, underwater deposition and marine disaster early warning. Traditional shallow sea bathymetry and underwater topography measurement require a lot of human and material resources, are time-consuming, and are limited in disputed sea areas.

Advances in the human exploration of deep space have resulted in increases in satellites and various types of remote sensing data. Recent research has focused on remote sensing bathymetry. Optical bathymetry is a passive remote sensing method; it is possible to obtain water depths based on differences in the transmission of light in different water bodies using multispectral data (including ultraviolet, visible, near-infrared, and mid-infrared wavelengths), providing a basis for estimating various water depth parameters [1,2]. Many

scholars [3–6] have focused on optical bathymetric remote sensing technology since the 1960s, developing various modeling methods for bathymetry inversion using multispectral satellite remote sensing data, including theoretical analytical models [7–9], semi-theoretical and semi-empirical models [10–14], and statistical models [15–18]. However, the application of these models is limited by the large number of optical parameters for the water body required in the model construction process. Alternatively, progress has been made in the development of semi-theoretical and semi-empirical models as well as statistical models. The CatBoost bathymetric inversion model improved the accuracy of multispectral bathymetric inversion to 1.09 m. Zhao et al. [19] concluded that machine learning algorithms are promising for optical bathymetric remote sensing based on analyses of five bathymetric simulation methods (i.e., deep learning, curve fitting, BP [back propagation] neural network, RBF [radial basis function] neural network, and SVM [support vector machine] neural network). Despite great success in optical experiments with different models, most of these studies have focused on shallow marine areas, such as rivers, lakes, and islands, with minimal atmospheric changes and good water quality. Because optical remote sensing images are affected by various environmental factors, the models are often only applicable to specific areas, with low universality.

Microwave radar bathymetry, mainly represented by synthetic aperture radar (SAR) bathymetry, is an active measurement technique with a high accuracy, wide coverage, high density of measurement points, short measurement period, low consumption, easy management, and high mobility [20]. In 1988, the "Seafloor Topography" research project involved a series of experimental studies using the Dutch digital aerial side-view radar (DDSLAR) system with X-band HH polarization in the sea > 30 km off the Dutch coast. Comparisons of radar images with digitized nautical charts revealed bright or dark streaks at the maximum slope of the terrain under different flow conditions [21]. Since then, SAR images have been used to obtain bathymetric information directly or indirectly. The bathymetry assessment system (BAS) developed by ARGOSS in the Netherlands is an example of an effective terrain detection system developed based on the AH model [22]. However, studies of bathymetry and underwater topography based on SAR images always involve computing waves, wave spectra, Fourier transform, etc. Recently, Huang et al. [23] refined a shallow sea topography detection method based on the wave characteristics of SAR images to reduce the error to 14.5%. Wang et al. [24] obtained simulated SAR images based on the 3D flow field model–radar simulation model; by continuously adjusting parameters to invert the bathymetry, a root mean square error (RMSE) of 3.4 m was obtained. For shallow marine areas with complex hydrographic conditions, wave spectra have been analyzed in bathymetry and underwater topography studies. Kaiguo et al. [25] applied an AH model and sea surface microwave scattering imaging simulation model, obtaining an RMSE value of 2.5 m, with an error of less than 10%. Yujin [26] used multiple time-series networks to analyze the shallow sand wave topography in Taiwan at scale and the inversion of the shallow water depth based on solar flare and multibeam data, with a mean square error of approximately 3.2 m. These studies apply complex inversion methods and yield average accuracies.

Deep learning, initially proposed in 2006, has emerging applications in a wide range of fields (e.g., biomedicine, smart cities, and aerospace research). Its outstanding data processing capabilities, such as image classification [27–29], target detection [30], and feature extraction [31], provide a novel processing method for water depth inversion. Pan et al. [32] proposed a bidirectional long- and short-term memory network (B-LSTM) to extract potential sequence features from multispectral images, obtaining a reduced error to within 0.6–1 m in the shallow sea (i.e., 0–12 m). Zhao et al. [15] proposed a BP neural network bathymetric inversion algorithm based on the elastic gradient descent model. The elastic gradient descent model directly changes the magnitude of the weight update value, eliminates the influence of the learning rate and momentum parameters, reduces computational parameters, quickly generates optimal model weights, and reduces the inversion error to 0.86 m for depths of 0–15 m. Style migration, a type of deep learning

introduced in 2015 [33], first addresses textural details and then combines component features for analysis so that the input content map has the style of the input style map. This algorithm is based on a convolutional neural network (CNN) for visual recognition, and Gupta et al. [34] evaluated four networks, InceptionV3, Resnet50, VGG16, and VGG19, and found that the VGG19 network is the most suitable for the style migration task. Chen et al. [35] applied the VGG19 model to study global deep sea bathymetry.

However, the above studies focused on areas within 20 m of clear water and deep sea areas. Relatively few studies have evaluated the application of deep learning to complex hydrographic environments and turbid waters. To this end, this paper proposes a novel bathymetric inversion method for turbid water in complex hydrographic environments in shallow seas based on the VGG19 framework for style migration. An innovative method is applied to minimize the distance conversion between back propagation by optimizing the loss function equation to ensure the uniformity of spatial resolution and spatial coverage area of the SAR image map and multibeam line map. The model with little loss, short inversion times, and high accuracy generates the image that best matches the style map. SAR image data and multibeam survey line data are used as the content map and style map, respectively, and point-to-point feature extraction learning was performed to obtain a bathymetric inversion map based on multibeam data with a high resolution and high accuracy for complex hydrographic turbid waters above 20 m.

The main contributions of this article are as follows.

(1) A new SAR image-based bathymetric inversion algorithm is proposed. In this algorithm, traditional preprocessing steps are used without considering the influence of natural factors, such as tides and wind fields, thereby addressing a limitation of SAR images and improving image utilization to a certain extent.

(2) Breaking the conventional focus on Fourier transform and wave spectrum analyses in topographic studies of the Taiwan Banks, we use a deep-learning-based algorithm for point-to-point learning, streamlining bathymetric inversion and relying more on machine learning for data, saving manpower and operating time.

(3) The loss function formula in the algorithm is innovatively optimized to introduce a formula that best fits the computational learning of the data in this experimental area, thus minimizing the propagation of the inverse distance conversion under the VGG19-based framework and establishing a precise shallow bathymetry inversion model able to adapt to the complex hydrographic environment of the open ocean.

(4) This experiment mainly studies the shallow seas of water depth less than 50 m. The accuracy of deep-learning-based bathymetry inversion in shallow waters with turbid water outside of 20 m was greatly refined. The error was reduced to 1.86 m in the water depth range of 20–40 m, and the relative error was only 6%, an improvement of 9.22% over previous estimates. The maximum correlation coefficient was 0.886 in the 20–30 m area, and the RMSE was 0.741 m, which was much better than those other studies for the same period.

## 2. Materials and Methods

### 2.1. Overall Framework of the Model

The basic process of neural networks is as follows: forward propagation, calculation of losses, backward propagation, and updating parameters. During the training process, the parameters are constantly changed, while the input data (i.e., training samples) are unchanged. In contrast, in the implementation of style migration, the network is already trained, and in the training, the parameters of the network do not change, but the input data change. The general flow is as follows.

(1) Unify the pixel resolution of the images to be generated.

(2) Find the same size content image (SAR preprocessed image map) and style image (multibeam line measurement image).

(3) Find a trained network. In the comparison of the effects of four networks (i.e., InceptionV3, Resnet50, VGG16, and VGG19), the VGG19 network had the best image style propagation effect [33] and was therefore used in this experiment.

In style migration models, network depth plays a crucial role. The VGG19 method is simple and satisfying in terms of network depth, using only $3 \times 3$ convolution and $2 \times 2$ pooling, yet it has 160 M parameters as well as 16 convolutional and 3 connection layers, allowing it to accommodate large amounts of data while extracting the most salient image features and reducing target loss. Typically, each layer in the network defines a nonlinear filter bank whose complexity increases with the location of that layer in the network. Figure 1 depicts the architecture of the CNN-based VGG19 pretraining model, where a color block represents a convolutional layer and a ReLU layer, with a pooling layer between the different blocks.

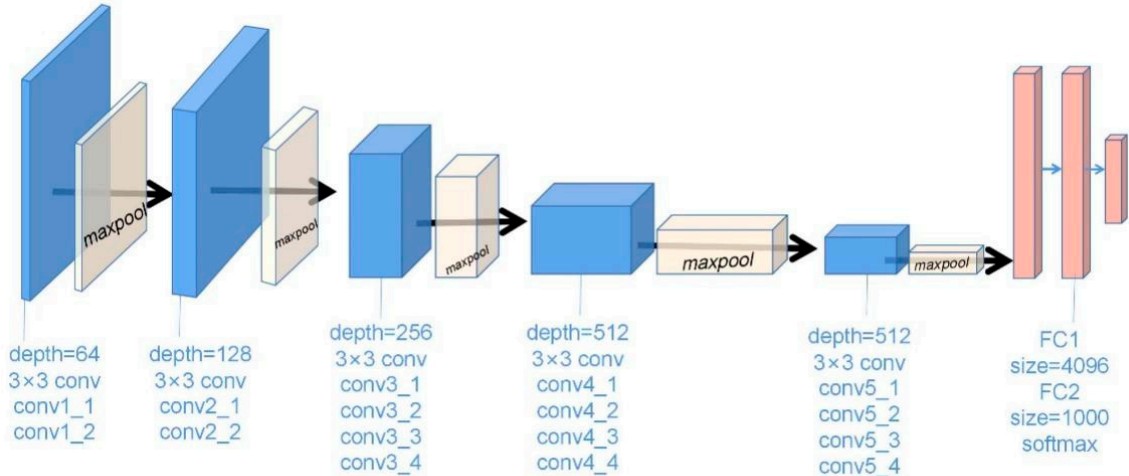

**Figure 1.** Architecture of the CNN-based VGG19 pretraining model.

(4) Create a random number of 3D arrays as the generated image.

(5) Drop the generated image (*G*), content image (*C*), and style image (*S*) into VGG19 and calculate the style loss and content loss.

(6) Implement back propagation (where only the gradient is propagated, and no parameters are updated) to obtain the loss about the gradient of the generated image.

(7) Update the data for the generated image using the optimization algorithm and repeat the last three steps.

An overview of the model process is shown in Figure 2. Because this experiment requires point-to-point learning of the pixel value of each image element point, the input image and output image are coded with latitude and longitude coordinate information in the code design part; accordingly, the output bathymetric inversion map is expressed as a grayscale map with latitude, longitude, and bathymetric values, which is convenient for accuracy measurements.

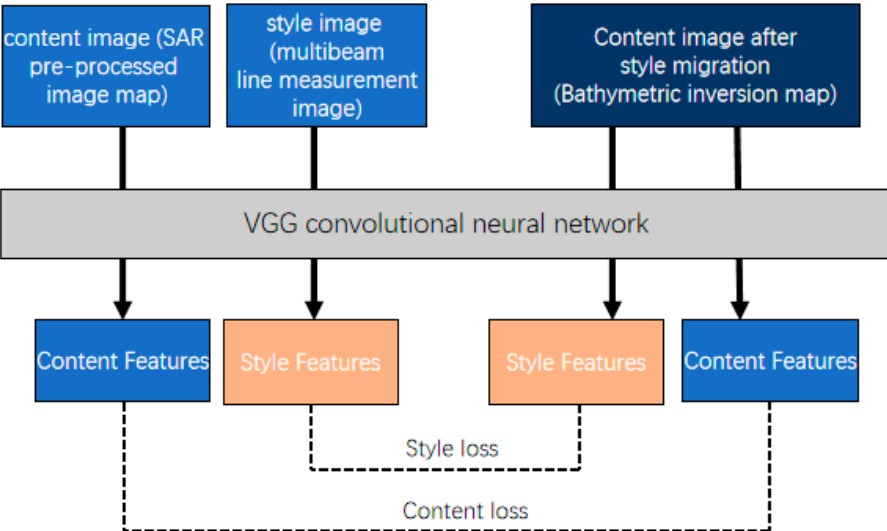

**Figure 2.** Overview of the model process.

### 2.2. Explanation of the Model Principle

The core optimization point is based on the previous style migration loss function. By continuously and repeatedly adjusting the distance loss, a formula is finally obtained that best meets the experimental requirements [36].

Overall loss: If the input is treated as a layer (Layer *G*), the output of the layer is its own parameters, and VGG19 is considered a loss function, resulting in the simplest gradient descent process in which the overall loss is divided into two parts: content loss and style loss.

$$Loss = \alpha L_{conent} + \beta L_{style} \tag{1}$$

where $\alpha$ and $\beta$ are two hyperparameters indicating the weights of these two losses, respectively, and need to be set and adjusted in the model itself.

A larger overall loss indicates a larger variance, and conversely, a smaller overall loss indicates a smaller variance.

Content loss: To calculate the content loss, a layer (e.g., the 7th ReLU layer) is selected in VGG19, and the output *G* for this layer regarding the SAR image map *C* and the inverse bathymetry map is calculated. The loss is calculated based on the above two output values. The content loss obtained with the most suitable formula for calculating the loss, i.e., the mean square error (MSE) loss function, is used to express and reduce the difference between two images, and simulated water depth combining SAR image data with multibeam data in the framework is generated to return to the model for learning. The formula is as follows.

$$L_{\text{conent}} = \frac{1}{2HWC} \sum_{ijk} \left( G_{ijk} - C_{ijk} \right)^2 \tag{2}$$

where *G* denotes the inverse bathymetric map, *C* denotes the simulated bathymetric map for the SAR image map, and *ij* denotes the pixel points for the input image representing the selected ReLU layer.

Because it is in a convolutional network, the output of ReLU is a four-dimensional tensor. In $C, G \in R^{N \times H \times W \times C}$, the four dimensions are the number of sample batches, height, width, and number of channels of feature values. Because only one generated image is optimized at a time after determining the content image and the style image, *N* is 1, and this dimension is omitted.

Style loss: The style loss is calculated for any layer by the same processing steps: calculate the output *S* for the layer about the style image and the output of the generated image *G*, calculate the Gram matrix *M* of the two output values, and calculate the mean square error loss for both Gram matrices.

The Gram matrix is calculated as follows:

$$M_{ij}^{G} = \sum_h \sum_w G_{\mathrm{hw}i} G_{hwj} \tag{3}$$

The style loss formula is as follows:

$$L_{\mathrm{style}} = \frac{1}{(2HWC)^2} \sum_i \sum_j \left( M_{ij}^{G} - M_{ij}^{S} \right)^2 \tag{4}$$

### 2.3. Accuracy Measurement

To assess the accuracy of bathymetric inversion, the correlation coefficient $R^2$ was used to express the correlation between the bathymetric inversion value and the true bathymetry value. Additionally, the RMSE was used to express the specific error value between the bathymetric inversion value and the true bathymetry value. The calculation formula is as follows.

$$RMSE = \sqrt{\frac{\sum_{i=1}^{n} \left( \hat{f}_i - y_i \right)^2}{n}} \tag{5}$$

$$R^2 = 1 - \frac{\sum_{i=1}^{n} (y_i - f_i)^2}{\sum_{i=1}^{n} (y_i - \overline{y})^2} \tag{6}$$

where $n$ denotes the number of data set values, $i$ denotes the sequence number for data set values, $f$ denotes the predicted value (the inverse value of water depth), and $y$ denotes the true value. In general, $R^2 > 0.95$ indicates a significant correlation, $R^2 \geq 0.8$ indicates a high correlation, $0.5 \leq R^2 < 0.8$ indicates a moderate correlation, $0.3 \leq R^2 < 0.5$ indicates a low correlation, and $R^2 < 0.3$ indicates a very weak relationship. The RMSE is the deviation between the predicted bathymetry value and the true value of bathymetry. It is very sensitive to extra-large or extra-small errors in a set of measurements; accordingly, it reflects the precision of the inversion.

### 2.4. Experimental Data

The Taiwan Banks are located near the southern exit of the Taiwan Strait, ranging from 22°30′N to 23°47′N and 117°17′E to 119°14′E, with a width of 60–80 km from north to south, a span of 150 km from east to west, and a total area of about 8800 km². The average water depth is about 20 m, with shallow water in the middle surrounded by gradually deepening water. The shallowest point is about 10 m, and the deepest point is in the southeast, reaching 64 m. The experimental data used in this paper are the non-full-coverage multibeam survey line data for the eastern shore of the Taiwan Banks, with a central latitude and longitude of 23°N and 118°E, The resolution is 5 m. Data were collected in the field on the Minlongvutu60828 cruise in 2012, and the ASF (search.asf.alaska.edu/) (accessed on 13 November 2022)acquired in the same year (Sentinel 1 16 m resolution SAR image data),as shown in Figure 3. The Multi-beam field survey equipment and accuracy are shown in Table 1. The validation data were full-coverage multibeam data collected in a hydrographic survey by the Second Institute of Oceanography, Ministry of Natural Resources, China, by the survey vessel *Zhejiayuke 001* in 2018. Because the original survey line data were non-full coverage, we used interpolation to fill in gaps, generating a complete set of multibeam data.

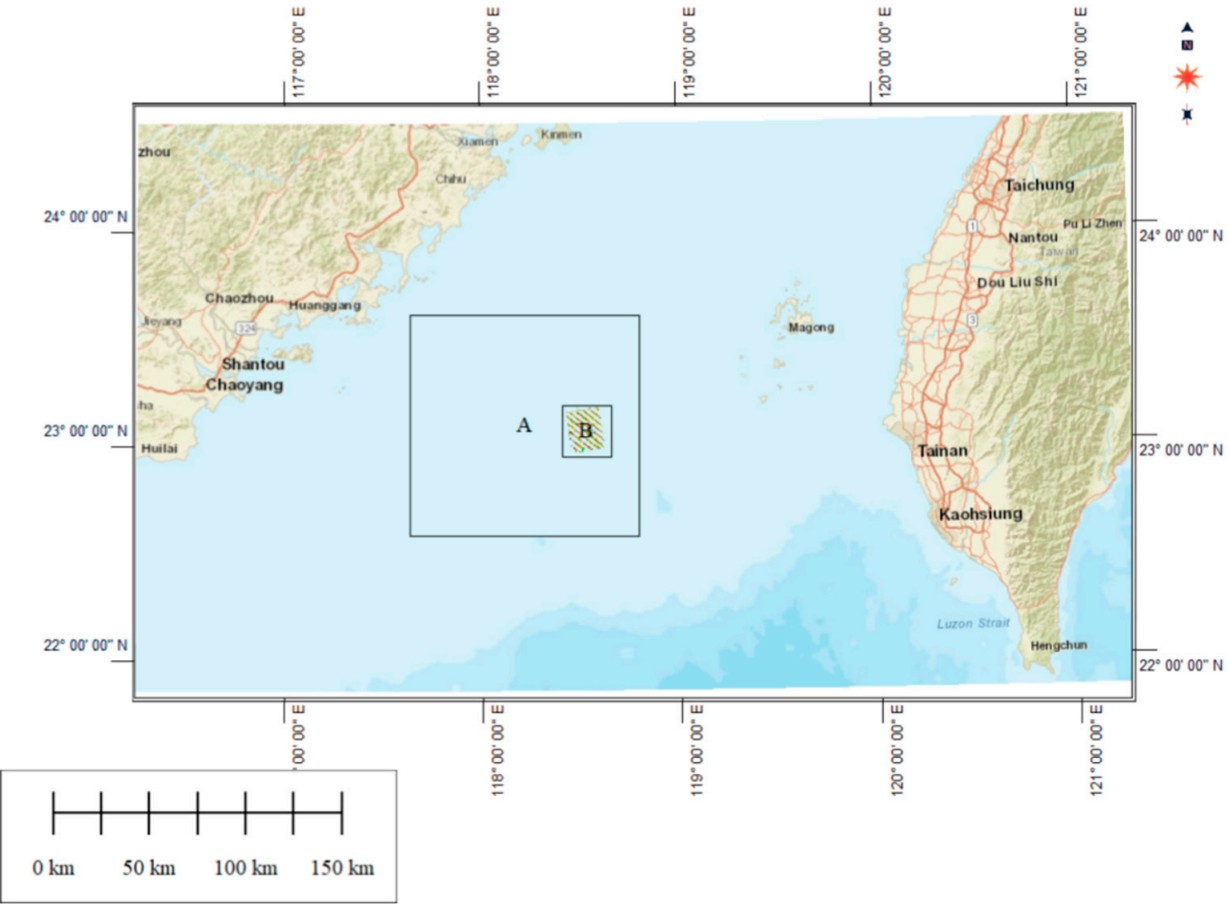

**Figure 3.** Schematic diagram of the Taiwan Banks location, where A is the panoramic position of SAR image data and B is the position of multibeam survey line data.

**Table 1.** Multi-beam field survey equipment and accuracy.

| Equipment Type | Instrument Type | Measuring Accuracy |
|---|---|---|
| Multibeam | R2SONIC 2024 | 6 mm |
| Position | Veripos | <0.1 m |
| Course | IXBlue OCTANS III | ±0.1° |
| Rise | IXBlue OCTANS III | ±5 cm |
| Sea level | RBR TD-2050 | Full range 5% |
| Surface sound velocity | AML MicroX | ±0.025 m/s |
| Speed of sound profile | RBR-420 CTD | ±0.5 m/s |

*2.5. Experimental Procedure*

2.5.1. SAR Image Data Pre-Processing

When the underwater flow field flows through the sand wave area with uneven terrain, the uneven terrain will cause the flow field to develop the phenomenon of evacuation and gathers, and the changes in the flow field will change the surface energy spectrum of the sea surface through wave–flow interactions. At the same time, under the joint action of the sea wind field, the roughness of the sea surface is produced. The changes in the roughness of the sea surface are reflected in the light of the sun's remote sensing image. Li et al. [37] found that the light and dark change in Taiwan's shallow beach was found in the Landsat-5 TM remote sensing image. Many years of studies have shown [26] that the depth of water and sea surface roughness in Taiwan are continuous and periodic. Although the changes in the two are different, the two show the same volatility cycle, which can be seen as a direct connection. In this experiment, we assumed that the backscatter coefficient of SAR

images was directly related to the water depth. Therefore, SAR images were preprocessed to obtain appropriate pictures. Thermal noise (which can be regarded as background noise) in SAR images will affect the accuracy of the estimated radar backscatter signal and was removed. Radiation calibration is the conversion of the received backscatter signal (energy) into a unitary (or unitless ratio) physical quantity, such as the backscatter coefficient. For SAR data, due to the penetrating nature of clouds, only radiometric calibration is required. Spot coherence is a common phenomenon in SAR images; it is treated as noise for various applications, such as SAR classification and as a valid signal for InSAR, etc. In this experiment, coherence was removed using the refined Lee filter (improved Lee filter) [38], which is the most commonly used coherence spot filter. It is an adaptive filter (the filter window can be automatically adjusted according to the region), and the processing effect is excellent. Topography correction is performed to reduce the extraneous error caused by topography offset in the inversion process. This yields the backscatter coefficient on a linear scale, which is usually a small positive value and needs to be converted to decibels for later data analysis and use. The decibelization (formula) has several advantages. First, the radar backscatter coefficients in decibels are close to the common Gaussian distribution. Second, after the decibelization of radar backscatter coefficients, the number of data storage bits can be smaller (can be stored from double precision double data to float floating point data), saving storage space. Third, after the decibelization of radar backscatter coefficients, visualization and data analysis are more convenient.

The decibelization equation is as follows.

$$dB = 10 \times \log_{10}(P/P_0) \tag{7}$$

where $P$ and $P_0$ denote the target and reference quantities, respectively. For the backscattering coefficient $\sigma_0$, the decibelization, in effect, performs the following logarithmic transformation.

$$\sigma_0(dB) = 10 \times \log_{10} \sigma_0 \tag{8}$$

We then obtained a more accurate backscattered map of the SAR image after pre-processing, and used this map as the content base map for bathymetric inversion. The preprocessed image was normalized to the line of the multibeam image, so that the spatial extent and resolution were the same, which is convenient for the later code to process the learning data.

### 2.5.2. VGGNET Computing Flow

This experimental model was inspired by style migration. The algorithm was improved under the large framework of pretrained VGG19, using the RMSE function as the loss function, which reduces data loss and substantially improves the inversion accuracy. The process involved one-to-one simulation inversion of the bathymetry values of image element points, extraction of inversion features, control of input data loss, and continuous gradient descent and optimization cycles. After iterative stabilization, a high-precision bathymetric inversion map with a resolution of 15 m was obtained, and the output includes an accurate latitude and longitude as well as bathymetric information without secondary processing.The flow chart of this experiment is shown in Figure 4.

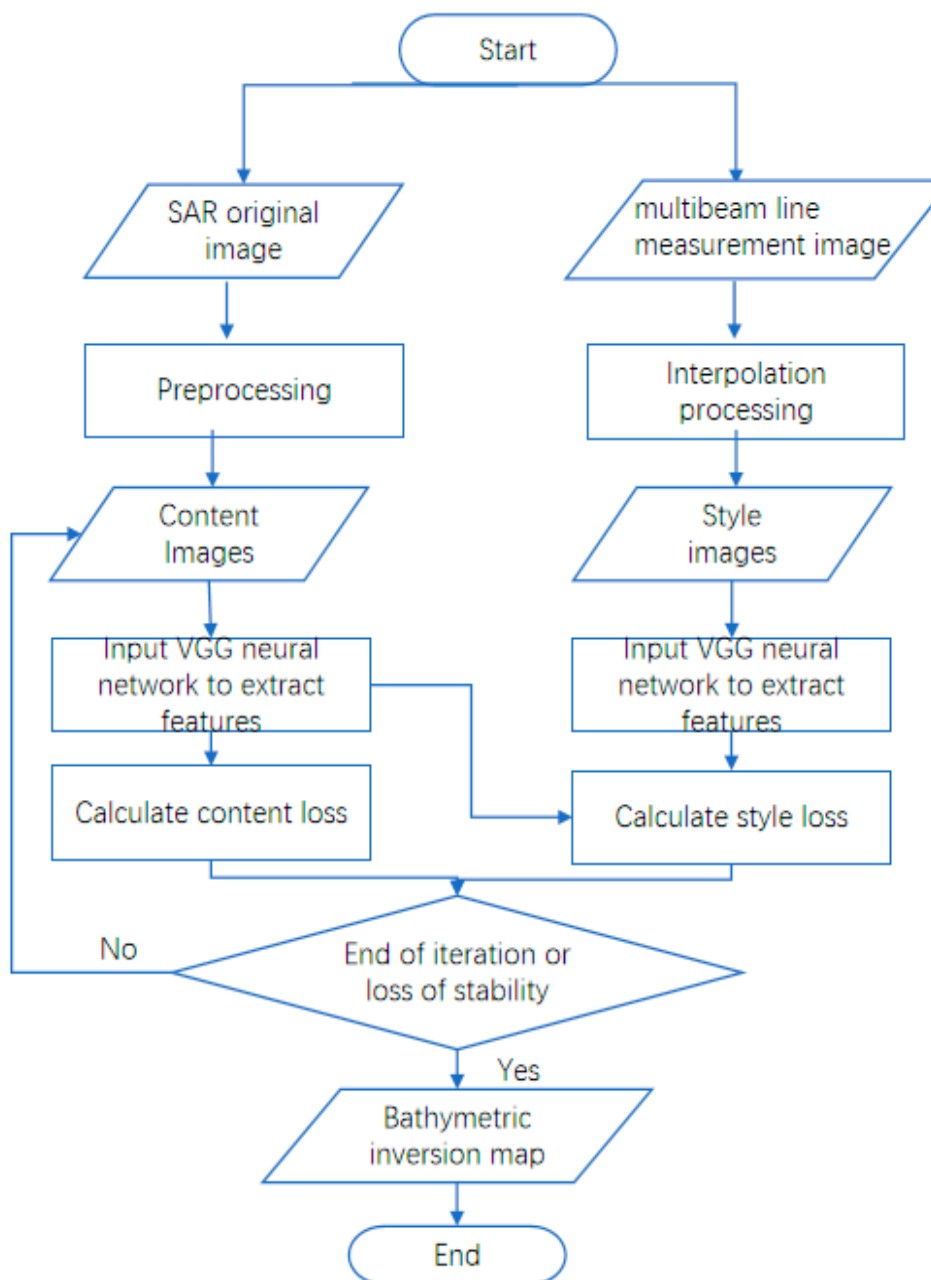

**Figure 4.** Flowchart of this experiment.

## 3. Results

The model was optimized for processed SAR image data as the content image and multibeam line interpolation data as the style image for style transfer learning. Finally, a multibeam fine-style-based bathymetric inversion map was output, as shown in Figure 5.

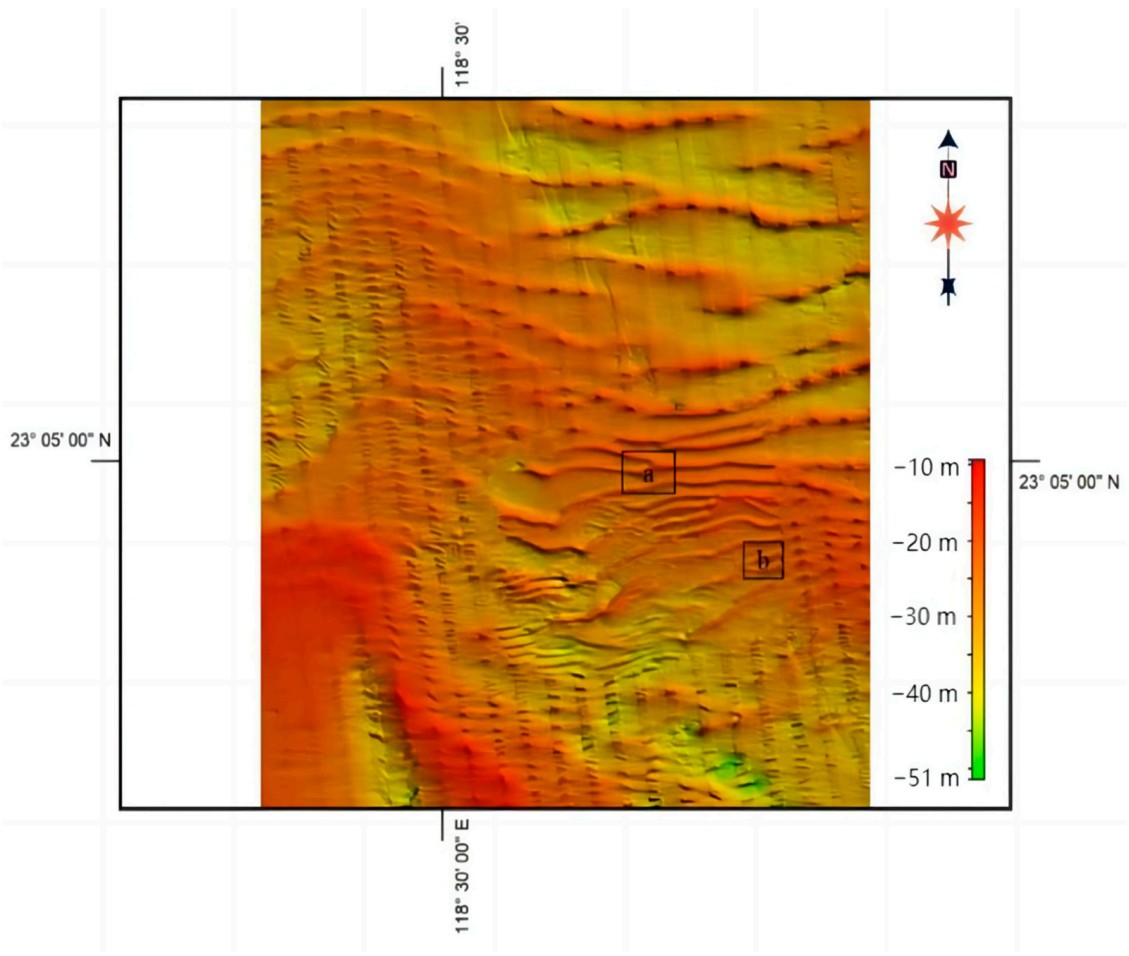

**Figure 5.** The 2012 bathymetric inversion map. a and b are the location of the interception area.

Two areas (a, b) in the inverse image are intercepted for display, as shown in Figures 6 and 7. The backscattering coefficient surface obtained from SAR images is finely inverted after learning the style of multibeam data, while the image roughness is optimized, as observed in the comparison. The output bathymetric inversion map is similar to the style of the measured line data and is accurate and clear. The images provide an intuitive tool for visual comparisons.

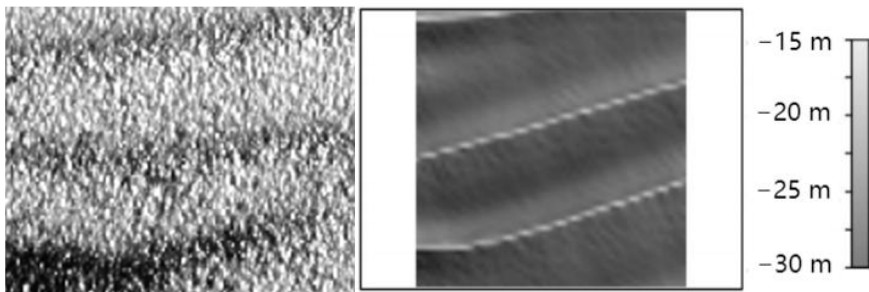

**Figure 6.** Original content image of area a (**left**) and water depth inversion image (**right**).

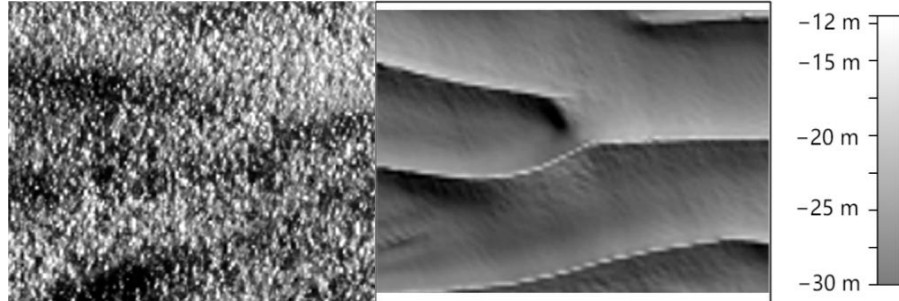

**Figure 7.** Original image of area b (**left**) and water depth inversion image (**right**).

Model stability is related to the loss function; the smaller the value of the loss function, the more stable the model and the better the fit to the data. As shown in Figure 8, the overall function loss in this experiment decreased linearly before iteration 2000 and remained stable after this inflection point at approximately 0. This indicates that after iteration 2000, the difference between the content image and the style image is small, and the pretrained VGG19 network fits well with the data from the Taiwan Banks. There was no overfitting, and convergence was always obtained, ensuring a sufficient data volume and reducing unnecessary data loss. Figure 9 represents the values of the loss function of the content map during the experiment. During the training process, the loss of the style map decreases, while the content map is corrupted and the loss increases; however, the overall loss decreases and remains at 500 after iteration 2000, indicating that the style map shares a similar style to that of the content map.

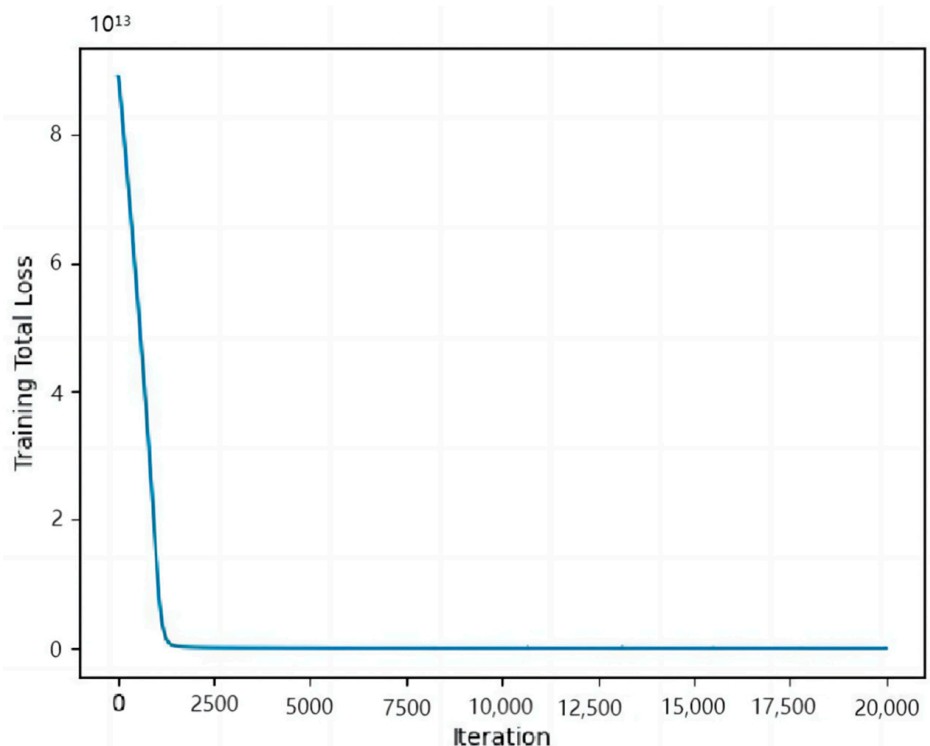

**Figure 8.** Overall loss function curve.

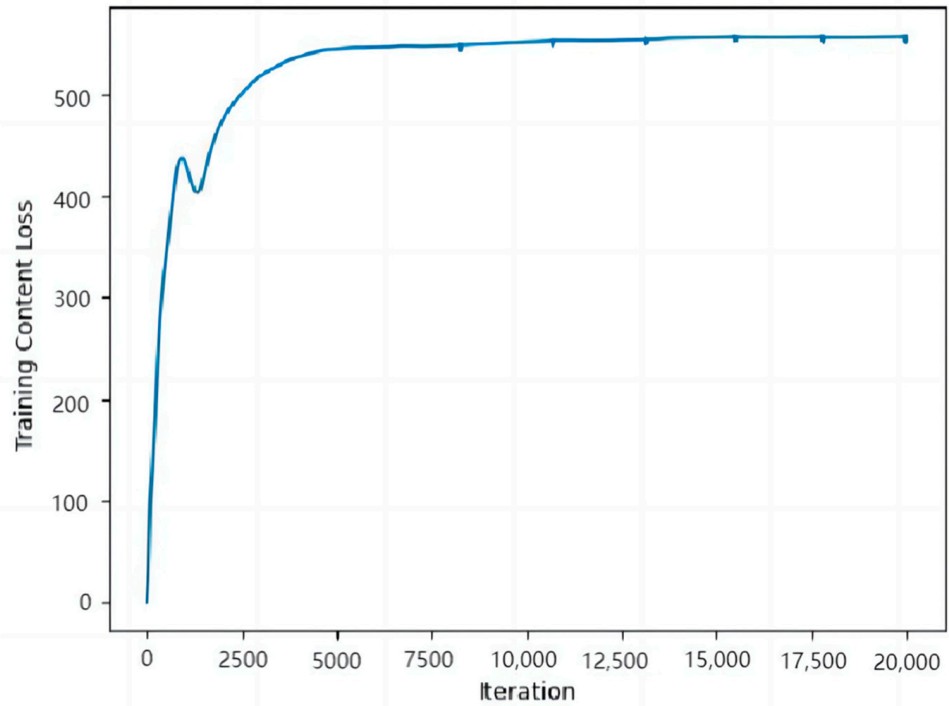

**Figure 9.** Content map loss function curve.

## 4. Discussion

The accuracy of the inversion results was expressed as the correlation coefficient $R^2$ and RMSE. The content image was changed to the 2018 SAR image data map to obtain the 2018 bathymetric inversion map (Figure 10) to verify the generalizability of the model. As shown in Figure 11, the data were almost linearly related, with low dispersion. There was a strong correlation ($R^2 = 0.8822$), indicating that the model showed good performance in the bathymetry inversion based on SAR images. The RMSE was 1.86 m. In the SAR inversion bathymetry model, wave spectral methods, such as Fourier transform, were not used; the bathymetry error was 1.86 m with a relative error of 6%, which was higher than those for the Fourier transform and other approaches in the interval above 20 m [39]. Relative error increased by 9.22% in the bathymetric correlation analysis in 2018 (Figure 12), the correlation coefficient $R^2$ decreased to 0.8026, and the bathymetric error increased to 2.45 m; there was still a strong correlation, suggesting that the decline in accuracy may be explained by the special environment of the shallows of the Taiwan Banks itself. The average bathymetry of the Taiwan Banks is around 20–30 m, and the average bathymetry of the study area was 28.6 m. The water is shallow, and the distribution of small and medium-sized sand waves is dense. The style data used in the experiment were based on measurements from 2012, which lead to the migration of small- and medium-sized sand waves in the study area, thereby decreasing the experimental accuracy. The inversion accuracy was reduced due to the influence of different SAR images.

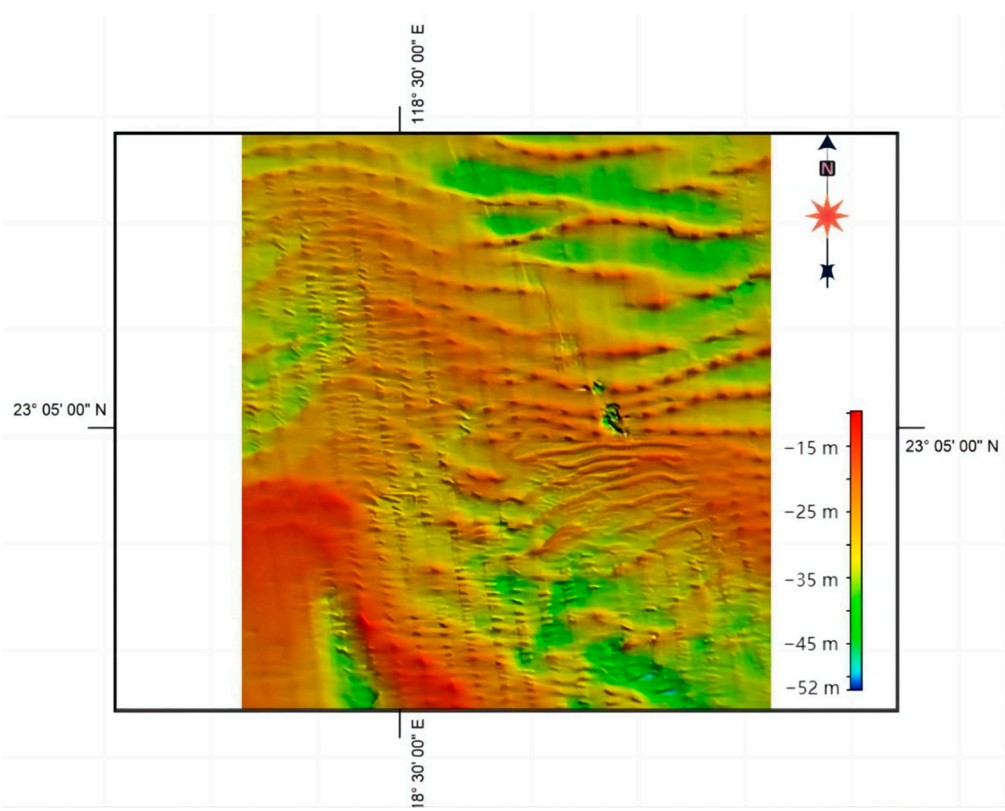

**Figure 10.** The 2018 bathymetric inversion map.

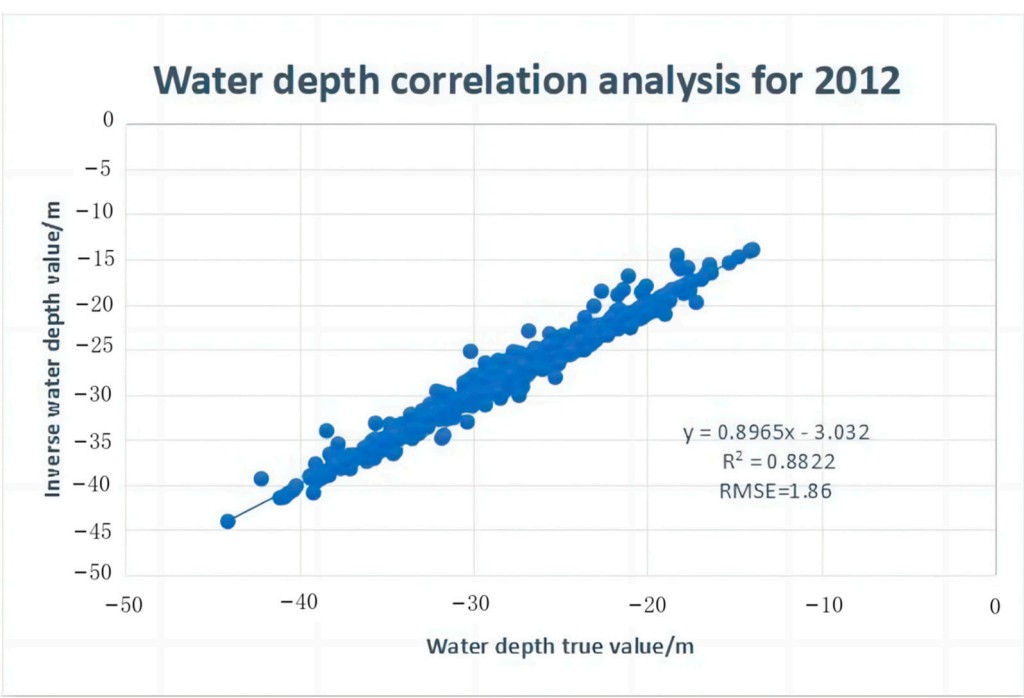

**Figure 11.** Water depth correlation analysis for 2012.

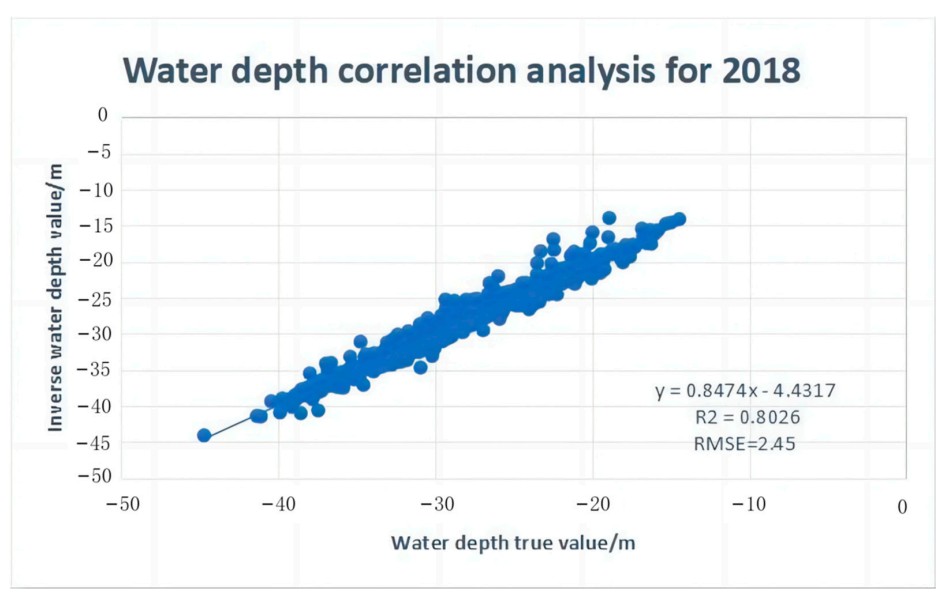

**Figure 12.** Water depth correlation analysis for 2018.

In addition, the minimum water depth in the study area was 13 m, and the maximum water depth was 40 m. The correlation coefficients between different water depth zones were analyzed, as shown in Figure 13. The highest correlation coefficient was 0.885, the lowest RMSE was 0.741 m in the water depth range of 20–30 m, and the lowest correlation coefficient was only 0.5237 in the range of 10–20 m. These findings indicate that for the Taiwan Banks, the model applicability is better for water depths above 20 m (representing the majority of the Taiwan Banks). The low accuracy of the inversion due to the small sample of data below 20 m is also a major factor contributing to this result. In addition, we found that images of the locations of sand ridges for the medium and large sand waves were unclear in the SAR image due to the shallow water depth, resulting in deviations in decibels and thereby reducing the accuracy of the inversion in the area below 20 m. However, in the range of 20–40 m, the model performed well, with strong correlations. The inversion effect was best at this interval, which greatly refines the accuracy of bathymetric inversion in the complex hydrographic environment above 20 m in the open sea.

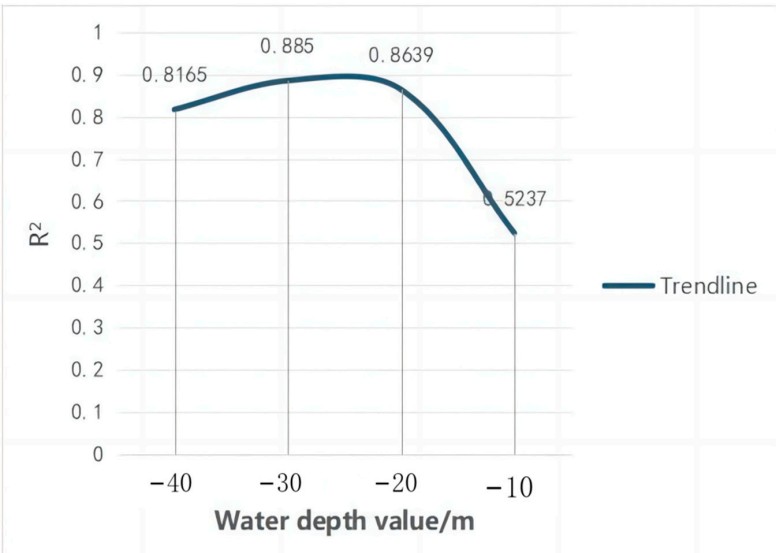

**Figure 13.** Trends in $R^2$ across different water depth intervals.

The true value and the inverse bathymetry value are determined differently, and the inverse bathymetry error map is obtained, as shown in Figure 14. The error map can visually and clearly display the error distribution of the inverse bathymetry, the error value, and the error density. The majority of points had error values distributed around 0 m, and there were significantly more deviated error values in the interval from –2 m to 0 m than from 0 m to 1 m. Thus, most of the inverse bathymetry values were 0–2 m less than the true values, and a few were about 0–1 m more than the true values. At the same time, the maximum error between the inverse bathymetry value and the true value was no more than 2 m, consistent with the RMSE estimate of 1.86 m, highlighting the credibility of the accuracy error of the bathymetry of the inverse of this model at approximately 1.86 m.

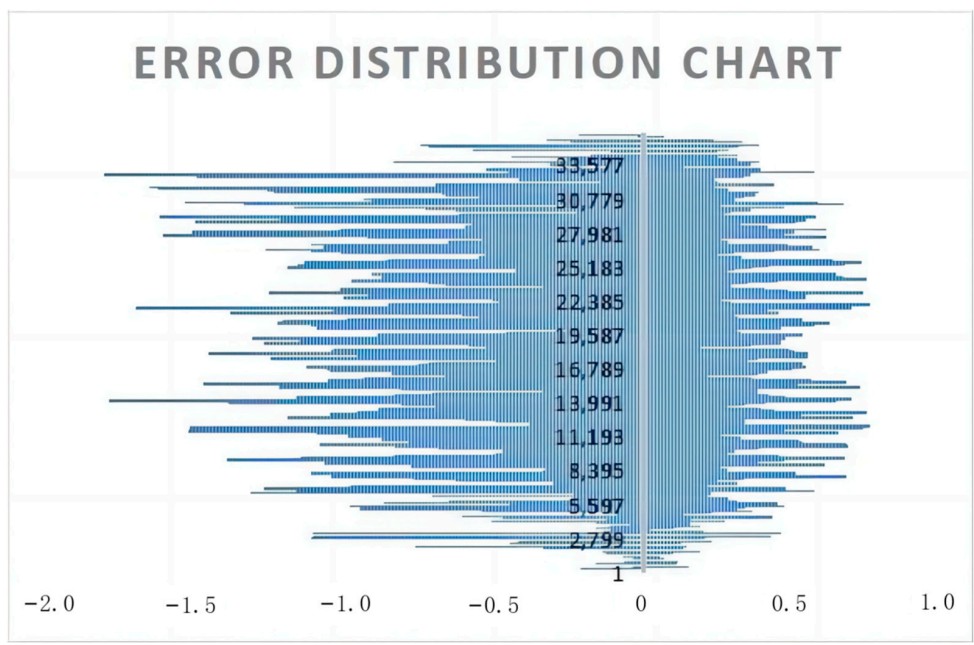

**Figure 14.** Error analysis chart.

## 5. Conclusions

Using a VGG19 network as the framework and an optimized style migration algorithm, this study proposes a new bathymetric inversion method based on SAR images and multibeam real-world data. This method allows SAR images to be used for bathymetry and underwater topography inversion after only traditional preprocessing steps, without considering the influence of natural factors, such as tides and wind fields, on imaging. This substantially reduces the difficulty of using SAR images and, to a certain extent, can improve image utilization. This algorithm is an improvement over conventional Fourier transform and wave spectrum analyses in topographic studies of the Taiwan Banks. Furthermore, it streamlines the steps involved in water depth inversion and utilizes machine learning to reduce manpower and operation time. We innovatively optimize the loss function formula in the algorithm and introduce a function formula that best matches the data calculation and learning in the experimental area, minimizing the propagation of the inverse distance conversion under the VGG19 framework. The high-precision shallow bathymetry inversion model can adapt to the complex hydrographic environment of the open sea, which benefits from low content loss, short inversion times, simple inversion steps, small error, and high accuracy. The model can be used to learn point-to-point from SAR images and multibeam data; therefore, low-resolution SAR images can adopt the detailed style of multibeam data to achieve inversion. During the experiment, the model remained stable, without anomalies, such as overfitting. Data set loss was minor, the results were accurate, and the process was efficient.

With respect to accuracy, $R^2$ was 0.8822, indicating a strong correlation between the inverse bathymetry value and the true value, and the RMSE was 1.86 m. Under the same bathymetry interval used for other methods of inversion in the Taiwan Banks, the error in this study was refined by 0.8 m in the range of 20–40 m, and the relative error was 6%, which was 9.22% better than that for other methods. To verify the generalizability of the model, we replaced the 2012 SAR image map with the 2018 image map and obtained an $R^2$ value of 0.8026, again indicating a strong correlation; however, the RMSE increased to 2.45 m, and the relative error increased by 3%. These increases may be explained by the migration of small- and medium-sized sand waves in the study area over time, causing misalignment and insufficient clarity for shallow water areas during SAR imaging. Analyses of the inversion accuracy over many years are needed to determine the universality of the model.

The study area was mostly located in the 20 m clear-water area. However, the correlation analysis for water depth intervals revealed that the model performs best in the 20–30 m interval, with a correlation coefficient as high as 0.886, with weaker inversion effects for intervals below 20 m. This error is directly related to the small number of water depth samples below 20 m in the experimental area. The number of samples determines the accuracy of the inversion. Accordingly, the model is suitable for the inversion of water depths in the shallow marine hydrographic environment in the interval of 20–40 m. The maximum correlation coefficient and RMSE were much better than previous estimates for the same period.

Inversion error is unavoidable. Although the model resolves large errors in the imaging process, there are still errors caused by the angle of incidence, imaging angle, multi-beam measurement process and data quality, and resampling and interpolation of line data [40–42]. These sources of error will have an impact on the final results, as they are uncontrollable. We can reduce the errors infinitely by optimizing the controllable algorithm, reducing human and material resources.

In future research, we expect to use the temporal variation in images to explore additional models that account for the migration of water depth and underwater topography over long time series. Additionally, we will improve the algorithm and increase the capacity to handle large data sets for bathymetric inversion over large areas, making the model applicable to multivariate data and expanding its scope of use.

**Author Contributions:** Conceptualization, X.L. and Z.W.; data curation, J.C., J.Z. and H.W.; formal analysis, J.C., X.L. and J.Z.; funding acquisition, X.L. and Z.W.; investigation, J.C., J.Z., H.W. and X.Q.; methodology, J.C., J.Z., H.W., X.C. and X.Q.; project administration, X.L. and Z.W.; resources, X.L. and Z.W.; software, J.C.; validation, J.C. and X.C.; visualization, J.C., H.W. and X.Q.; writing—original draft, J.C.; writing—review and editing, J.C. and X.L. All authors have read and agreed to the published version of the manuscript.

**Funding:** This study is supported by the National Natural Science Foundation of China (41830540, 42006073), Research Fund of the Second Institute of Oceanography, Ministry of Natural Resources (JG2101), the Oceanic Interdisciplinary Program of Shanghai JiaoTong University (SL2020ZD204, SL2004), Natural Science Foundation of Zhejiang Province (LY23D060007, LY21D060002), National Key Research and Development Program of China (2022YFC2806600, 2020YFC1521700, 2020YFC1521705 and 2022YFC3003803), The Open Fund of the East China Coastal Field Scientific Observation and Research Station of the Ministry of Natural Resources (ORSECCZ2022104), Zhejiang Provincial Project (330000210130313013006).

**Data Availability Statement:** The authors would like to express gratitude to ASF(NASA Distributed Active Archive Center) and Key Laboratory of Submarine Geosciences, Second Institute of Oceanography, Ministry of Natural Resources for data services.

**Acknowledgments:** My deepest gratitude goes first and foremost to Luo Xiaowen, my tutor, for his constant encouragement and guidance. He has walked me through all the stages of the writing of this paper Without his consistent and illuminating instruction, this paper could not have reached its present form. Second, I would like to express my heartfelt gratitude to Wu Ziyin, Zhou Jieqiong, Wan Hongyang, Qin Xiaoming and Chen Xiaolun, who have instructed and helped me a lot in the past two years. Last my thanks would go to my beloved family for their loving considerations and

great confidence in me all through these years. I also owe my sincere gratitude to my friends and my fellow classmates who gave me their help and time in listening to me and helping me work out my problems during the difficult course of the paper. I would like to express my gratitude to all those who helped me during the writing of this paper.

**Conflicts of Interest:** The authors declare no conflict of interest.

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
