# Peer review of "High-Precision Inversion of Shallow Bathymetry under Complex Hydrographic Conditions Using VGG19—A Case Study of the Taiwan Banks"

_remotesensing, doi:10.3390/rs15051257_

Round 1

Reviewer 1 Report

This work considers the application of Convolutional Neural Network (CNN) VGG19 for the assessment of sea bathymetry frome Synthetic Aperture Radar (SAR) image and multibeam data. The work was performed on the shallow water of Taiwan. I have some suggestions that in my opinion must be considered for the improvement and pubblication of this work.

I strongly suggest to insert a map showing the multibeam bathymetry and inversion bathymetry map, because these are not reported anywhere in the main text.

Then, in the Introduction Section, a definition of shallow sea bathymetry must be provided. This is important to understand the depth range that can be analyzed through the VGG19 and SAR images. For example the depth ranges reported in the main text seem to be referred to deep water (e.g. 20 - 30 m).

Furthermore, before the desciption of framework of the model (subsection 2.1), a subsection with bathymetric data must be inserted. This is important for a discussion of the kind of data acquired.

In the section 3, the considerations concerning the detection of sand waves must be better discussed. For example, the ridges detected from SAR images could be relative to the wave crest or roughness of sea surface. Please insert a discussion about this aspect.

Other comments were highlighted in the attached pdf file.

Kind regards.

Reviewer 2 Report

This paper presents a new method of high-precision inversion of shallow bathymetry for synthetic aperture radar (SAR) based on VGG19 neural network. The author uses VGG19 neural network to realize the transfer learning of the measured data from the synthetic aperture radar image style to the multibeam image style, and successfully integrates the advantages of the synthetic aperture radar, such as wide coverage, high measurement density, rapidity and the advantage of high accuracy of multibeam measurement. It is verified that the correlation between the inversion data and measured data is strong, and the measurement accuracy is also improved compared with previous studies.

        But it still remains some problems:

(a)    In figure 7, the training total loss decreased rapidly and remained approximately 0 after iteration 2000, this seems to mean the model is in an over-fitting state, how to understand it?

(b)   In figure 9 and 10, this paper used the data of 2012 to train the model, and the data of 2018 for validation. R2 changed from 0.8822 to 0.8026, which decreased rapidly. Should we add more year data as training sets to enhance the stability of the model?

(c)    In figure 11, when the water depth is under 20m, the R2 decreased rapidly from 0.8639 to 0.5237, it means this model’s poor applicability in shallow water, how to understand and deal with it?

(d)   The part 1 introduction and part 2 methodology and data are too long.

(e)    In the row 323 page 11 of this paper, there is a citation error. The author seems to explain figure 7 while writing “as shown in figure 6”.

Round 2

Reviewer 1 Report

The paper was improve as suggested in my comments.

Kind regards.